# IoT-Based System for Real-Time Monitoring and AI-Driven Energy Consumption Prediction in Fresh Fruit and Vegetable Transportation

**DOI:** 10.3390/s25247475

**Published:** 2025-12-09

**Authors:** Chayapol Kamyod, Sujitra Arwatchananukul, Nattapol Aunsri, Rattapon Saengrayap, Khemapat Tontiwattanakul, Chureerat Prahsarn, Tatiya Trongsatitkul, Ladawan Lerslerwong, Pramod Mahajan, Cheong-Ghil Kim, Di Wu, Saowapa Chaiwong

**Affiliations:** 1School of Applied Digital Technology, Mae Fah Luang University, Chiang Rai 57100, Thailand; chayapol.kam@mfu.ac.th (C.K.); sujitra.arw@mfu.ac.th (S.A.); nattapol.aun@mfu.ac.th (N.A.); 2Computer and Communication Engineering for Capacity Building Research Center, Mae Fah Luang University, Chiang Rai 57100, Thailand; 3School of Agro-Industry, Mae Fah Luang University, Chiang Rai 57100, Thailand; rattapon.sae@mfu.ac.th; 4Integrated AgriTech Ecosystem Research Group, Mae Fah Luang University, Chiang Rai 57100, Thailand; 5Department of Mechanical and Aerospace Engineering, King Mongkut’s University of Technology, North Bangkok 18000, Thailand; khemapat.t@eng.kmutnb.ac.th; 6National Metal and Materials Technology Center, National Science and Technology Development Agency, Pathumthani 12120, Thailand; chureerp@mtec.or.th; 7School of Polymer Engineering, Suranaree University of Technology, Nakhon Ratchasima 30000, Thailand; tatiya@sut.ac.th; 8Agricultural Innovation and Management Division, Faculty of Natural Resources, Prince of Songkla University, Songkhla 90110, Thailand; ladawan.l@psu.ac.th; 9Department of Systems Process Engineering, Leibniz Institute for Agricultural Engineering and Bioeconomy (ATB), 14469 Potsdam, Germany; pmahajan@atb-potsdam.de; 10Department of Computer Science, Namseoul University, Cheonan-si 31020, Republic of Korea; cgkim@nsu.ac.kr; 11College of Agriculture and Biotechnology, Zhejiang University, Hangzhou 310058, China; di_wu@zju.edu.cn

**Keywords:** battery, Boosting Machine, GPS, LiFePO_4_, model, perishable logistics, temperature

## Abstract

Temperature and humidity excursions during transport accelerate quality loss in fresh produce. This study develops and validates a self-contained Internet of Things (IoT) platform for in-transit monitoring and energy-aware operation. The battery-powered device operates independently of vehicle power and continuously logs temperature, relative humidity, GPS position, and onboard power draw. Power budgeting combines firmware-level deep-sleep scheduling with a LiFePO_4_ battery pack, enabling uninterrupted operation for up to four days. Using ∼10,000 time-stamped observations collected over four consecutive days in a standard dry truck (non-commercial validation), we trained and compared Gradient Boosting Machine (GBM), Random Forest (RF), and Linear Regression (LR) models to predict energy consumption under varying environmental and routing conditions. GBM and LR achieved high explanatory power ($R^{2}\approx 0.88$) with a mean absolute error of 0.77 A·h, while RF provided interpretable feature importance data, identifying temperature as the dominant driver, followed by trip duration and humidity. The end-to-end system integrates an EMQX MQTT broker, a Laravel web application, MongoDB storage, and Node-RED flows for real-time dashboards and multi-day historical analytics. The proposed platform supports proactive decision-making in perishable logistics, with the AI analysis validating that the collected time-aligned on-route data can configure sampling/transmit cadence to preserve autonomy under stressful conditions.

## 1. Introduction

Fresh fruits and vegetables are highly susceptible to temperature and relative humidity (RH) excursions during transport; even brief spikes accelerate senescence, moisture loss, and microbial activity, shortening their shelf life and eroding their value. Passive interventions—such as pre-cooling and composite thermal insulation—can mitigate heat load, as demonstrated in postharvest studies of okra and related produce [1,2,3]. Yet, passive measures do not detect on-route fluctuations, adapt to events, or provide traceability at the granularity required for modern logistics, indicating the need for a self-contained Internet of Things (IoT) platform that combines continuous sensing with energy-aware operation and analytics, designed to function reliably without tapping vehicle power.

### Related Studies

This subsection reviews prior research on the use of IoT for agriculture and food logistics, offline data logging with integrity, supply-chain analytics, and edge/AI systems for energy-aware operation, and highlights the gaps in the literature that motivated the proposal of our system.

Over the past decade, IoT has matured from pilots to large-scale systems employed in field sensing, networked telemetry, and cloud/edge analytics. Surveys and scoping studies demonstrate its benefits in terms of productivity gains, labor reduction, and decision support in agriculture, but also emphasize hurdles in its adoption, interoperability, and governance [4,5,6,7,8,9,10,11,12]. Within agri-food logistics, reference architectures and domain syntheses focus on end-to-end traceability and near-real-time decision support that integrate sensor streams with logistics information systems [13,14,15,16,17,18]. This study supports a design that binds environmental telemetry to route and time contexts (GPS, duration) and implements it in operator dashboards.

Transport corridors often experience coverage gaps; so, robust local logging with authenticated backfill is as critical as live streaming. A recent study by Garrido-López et al. [19] demonstrates the practical implementation of cellular IoT data loggers for monitoring perishable goods, achieving approximately one month of battery autonomy through optimized power consumption strategies in NB-IoT-enabled devices validated across refrigerated-transport routes. Studies on secure loggers, integrity checking, and connected-vehicle integration highlight patterns in the adoption of microSD persistence, periodic synchronization, and verifiable transfers to the cloud [20,21,22,23,24]. These design cues indicate the need for an offline-first workflow with device-scoped topics and access control so that records remain continuous across signal dropouts [25,26,27,28,29,30,31].

Studies across supply-chain analytics, inventory tracking, and forecasting show how continuous environmental data improves resource allocation and quality control [32,33,34,35,36,37,38,39,40,41,42,43,44]. Complementary studies on IoT logistics and fleet systems (e.g., context-aware transport, roadside units, vehicle telemetry) demonstrate gains in visibility and efficiency but typically assume stable power or infrastructure [45,46,47,48,49,50,51,52,53,54,55,56,57,58].

Processing closer to the source reduces latency and uplink dependence, enabling resilient, responsive monitoring in the field [59]. In parallel, reviews and applications of AI in agriculture and agri-food logistics show that lightweight, interpretable models can deliver practical value on modest datasets, particularly for resource-constrained systems [13,60]. Recent advances in ensemble modeling for cold-chain applications demonstrate improved prediction accuracy; for example, Luo et al. [61] reported that a K-medoids + long short-term memory (LSTM) + XGBoost ensemble achieved a mean absolute error of 2.5343 for temperature prediction in agricultural cold-chain loading environments, demonstrating the value of hybrid approaches for thermal dynamics modeling. Taken together, these ideas indicate the need for edge-centric designs in which firmware power policies and ML prediction co-evolve.

Beyond agriculture and logistics, operational energy forecasting has been investigated in several domains, including emergency responses, smart homes, manufacturing, and healthcare facilities. A recent study on time-series methods for use in emergency contexts highlights the role of short-horizon forecasting in supporting decision-making during critical events [62]. Actor–critic energy management schemes for smart homes use predictive models to schedule loads under dynamic tariffs [63], while deep learning has been applied to energy forecasting and condition monitoring in the manufacturing sector [64], and systematic reviews have synthesized AI-based energy prediction in healthcare facilities [65]. These cross-domain studies reinforce the value of compact, operational energy predictors and motivate our use of lightweight models for battery budgeting in perishable logistics.

Despite progress in this field, two practical solutions for perishable logistics need to be established: (i) *stand-alone autonomy*—battery-operated devices that operate reliably across multi-day domestic routes without vehicle power; (ii) *operationalized energy analytics*—models that explain and predict power draw as conditions change so that sampling, transmission, and battery sizing can be tuned proactively. Existing IoT deployments in agriculture and transport seldom integrate these capabilities within one deployable platform [1,2,3,4,5,6,7,8,9,10,11,12,13,14,15,16,17,18,20,21,22,23,24,25,26,27,28,29,30,31,32,33,34,35,36,37,38,39,40,41,42,43,44,45,46,47,48,49,50,51,52,53,54,55,56,57,58,59,60]. While progress has been made in IoT power management, rahman et al. [66] demonstrate that hybrid long short-term memory (LSTM) and gated recurrent unit (GRU) architectures can achieve effective power forecasting in IoT environmental monitoring systems (R^2^ 82.04%, MAE 3.78%), providing a foundation for energy-aware operation in resource-constrained deployments.

Device-level studies on perishable goods/food transport generally emphasize traceability and continuous cloud visibility, often assuming vehicle power or cellular backhaul. For example, Pal and Kant [16] outline an IoT sensing infrastructure for fresh-food supply chains that relies on gateways and stable connectivity, while Bhutta and Ahmad [57] focus on secure identification and real-time cloud updates via cellular links. In contrast, our design is a **vehicle-independent, battery-first logger** that fuses Temp/RH, GPS, and onboard energy; operates **in real-time and on-device** with **near-real-time** cloud updates when in range; and guarantees **verified backfill** across coverage gaps encountered on domestic dry-truck routes. To address these gaps, we develop a battery-powered, ESP32-based data logger that continuously records temperature, RH, GPS, and onboard power draw; implements firmware-level deep-sleep scheduling; and sustains multi-day autonomy using a LiFePO_4_ pack. In deployment, the logger connects via an 802.11 network to a **battery-powered cellular (4G-SIM) router** for WAN access, enabling near-real-time cloud updates when coverage exists, and authenticated backfill otherwise; multiple nodes can share the same SSID while publishing to device-scoped topics with separate credentials. Using ∼10,000 time-stamped observations collected over four transport days, we train and compare Linear Regression (LR), Gradient Boosting Machine (GBM), and Random Forest (RF) models to predict energy consumption under changing environmental and routing conditions. In our experiments, LR and GBM provide high explanatory power ($R^{2}\approx 0.88$), while RF yields comparable errors and interpretable feature importance (temperature > duration > humidity), informing our choice of firmware policy and battery size. The end-to-end stack integrates EMQX (MQTT), a Laravel web application, MongoDB, and Node-RED for real-time dashboards and multi-day historical analytics. Distinct from prior traceability-focused deployments, our design couples vehicle-independent uplink with authenticated backfill and operationally uses short-horizon energy prediction to guarantee a ≥3-day autonomy target (4 days of autonomy were achieved) without vehicle power.

## 2. Development of the IoT-Based Monitoring System

### 2.1. System Architecture

The platform follows a two-package layout to simplify installation and ensure electrical isolation. *Packet I* houses a sensing and compute system: an ESP32 microcontroller with Wi-Fi/Bluetooth, a calibrated temperature/RH sensor, a GPS module, a real-time clock (RTC), and a microSD card for resilient local logging. *Packet II* contains power and connectivity: a LiFePO_4_ battery pack with protection, DC–DC conversion (12 V and 5 V rails), and a Wi-Fi modem. The units are coupled via a keyed harness with over-current protection. Environmental measurements and GPS are timestamped on-device and published via MQTT when a link is available; otherwise, records accumulate locally and are backfilled upon reconnection. A web application renders live and historical views with user-level access controls. Figure 1 presents the system architecture. The components of this system are as follows:

### 2.2. Designed Hardware

The ESP32 orchestrates sampling, storage, and transmission. The SHT-class digital sensor provides temperature (–40 °C to +125 °C) and 0–100% RH coverage. A NEO-6M GPS module reports position and speed to indicate route context and travel duration. The RTC maintains stable timekeeping throughout deep-sleep cycles, and a level-shifted microSD interface (SPI) provides sufficient storage during coverage gaps. The power stage employs a LiFePO_4_ pack with an adequate size for multi-day autonomy, with conversion efficiency and thermal limits accounted for in the budget. All connectors are vibration-rated, and EMI/ESD design rules are observed to protect the MCU and sensor lines. The design of the controller board is shown in Figure 2, Figure 3 and Figure 4. The sensing head is mounted near the cargo midline at approximately a 1.5 m height, away from walls and door gaps to reduce boundary-layer effects; multi-node layouts for spatial gradients will be addressed in future research.

The controller is an **ESP32-WROOM32** (Espressif Systems, Shanghai, China)(dual-core MCU with integrated 802.11 b/g/n), and temperature and RH are measured by an **SHT3x-class** digital sensor (I^2^C; –40 °C to +125 °C; 0–100 %RH; factory calibration). A u-blox **NEO-6M** GPS module (UART, 1 Hz NMEA) provides position and speed, while a **DS3231** RTC (I^2^C; ±2 ppm) maintains timekeeping throughout deep sleep. Local persistence uses a microSD card (SPI, FAT32) with CSV records of the form {timestamp, seq_id, *T*, RH, lat, lon, speed, energy, CRC16}. For WAN access, the device connects over an 802.11 network to a **battery-powered 4G-SIM router** (vehicle-independent uplink) and publishes via *MQTT over TCP* (EMQX) to *device-scoped topics* with per-device credentials, *QoS 1*, and retained configuration; records carry a monotonic seq_id and CRC16, and the backend performs idempotent inserts keyed by (device, seq_id). Multiple sensor nodes can share the same SSID and publish concurrently under separate topics.

### 2.3. Communication and Data Flow

An EMQX MQTT broker mediates device topics, and a Laravel web application exposes role-based dashboards and APIs; MongoDB stores time-series documents for multi-day analysis, while Node-RED flows drive real-time visualization and alerting. Devices connect via an **802.11 network to a battery-powered 4G-SIM router** (vehicle-independent uplink) and publish via *MQTT over TCP* to *device-scoped topics* with per-device credentials and *retained configuration*. Payloads carry a monotonic seq_id and CRC16; the backend performs *idempotent inserts* keyed by (device, seq_id) to prevent duplication. When WAN coverage drops, the device continues to perform real-time on-device sensing and logs measurements to the microSD; on reconnection, it *replays buffered records in order* for *authenticated backfill*. Multiple sensor nodes can share the same SSID and publish concurrently under separate topics. For fast dashboard rendering, latitude and longitude are rounded in the display, while the stored time series and CSV exports retain the precise coordinates. This opportunistic MQTT pattern (real-time on-device; near-real-time to cloud when in range) maintains visibility across the route and yields a synchronized stream used to validate simple configuration rules for sampling and transmit cadence.

## 3. Materials and Methods

The proposed platform targets domestic-produce transport using low-cost dry trucks. Across four consecutive days, the device logged temperature (°C), RH (%), GPS position/speed, RTC timestamps, and onboard power draw data to the microSD, uploading the data via MQTT opportunistically. Field trials were conducted in a standard dry truck over representative urban–peri-urban corridors to validate system reliability, online/offline continuity, and multi-day autonomy. Live, commercial produce shipments were *not* included at this stage; the goal was to exercise sensing, logging, and backhaul under realistic motion/coverage conditions prior to goods-specific trials.

*Data schema and precision*. The CSV records include precise latitude/longitude (6 decimal places; last 4–5 digits vary by fix), e.g., ..., 19.907263, 99.832147, .... To ensure UI responsiveness, the dashboard displays rounded coordinates (to two decimal places), while the backend and exports retain the precise coordinates.

### 3.1. Dataset and Preprocessing

Time stamps were aligned at 1 Hz. Short gaps (≤5 s) caused by mobility and handovers were linearly interpolated, while longer gaps remained (no imputation) and were backfilled to the cloud upon reconnection. To suppress transition spikes around lab→vehicle handovers, we applied an interquartile-range (IQR) filter with a Tukey multiplier k=1.5, computed independently each day day and for each device. No additional smoothing was performed. Features were scaled (min–max) with parameters fitted on the training folds only and applied to the held-out test. For fast dashboard rendering, latitude and longitude were rounded in the display, while the stored time series and CSV exports retained the precise coordinates. Over the four days, daily record counts were approximately balanced at ∼2400–2700 per day (total ∼10,000).

### 3.2. Train/Test Split, Cross-Validation, and Metrics

A chronological 80/20 split was used to emulate forward-looking generalization. Model selection relied on 5-fold cross-validation within the training window. We report the mean absolute error (MAE), mean squared error (MSE), and coefficient of determination ($R^{2}$) for the held-out window; to reduce route-specific leakage, cross-validated selections were reapplied to the held-out split.

### 3.3. Models and Hyperparameters

We evaluated the following lightweight regressors suitable for edge-constrained telemetry: Linear Regression (LR), Gradient Boosting Machine (GBM), Random Forest (RF), *k*-Nearest Neighbors (*k*-NN), Support Vector Machine (SVM, RBF kernel), and a small multilayer perceptron (MLP). Five of these (LR, SVM, k-NN, GBM, and MLP) represent the main candidate models in the aggregate comparison plot (Figure 5), while RF is used primarily as an interpretable ensemble to provide feature importance (Figure 6) and predicted-vs-actual data (Figure 7); its performance metrics are reported in Section 4.2.

#### 3.3.1. Feature Set

The predictors comprise temperature (°C), relative humidity (%), route duration (elapsed minutes), and *transmission activity*, with the latter defined as the count of MQTT publications in the trailing 60 min window (continuous). No interaction terms were included in the main models, as the focus is on operational prediction.

#### 3.3.2. Configuration

The representative settings were as follows: RF (100–200 trees, max depth 10–20, minimum split 2–5); GBM (learning rate 0.1, 100–200 estimators); SVM (RBF, C=1, γ=scale); *k*-NN (k=5); MLP (two hidden layers, 64–128 units, ReLU, Adam). The grid search was nested within the cross-validation.

*Rationale and energy-awareness:* This subsection consolidates the above information regarding “Energy-Aware Analytics”: The the models are trained to predict power draw as a function of ambient temperature, RH, route duration, and transmission activity, with the aim of informing firmware duty-cycling and battery sizing. The performance metrics and feature importance for the trained models are reported and interpreted in Section 4, where we also discuss how they feed back into sampling cadence, radio coalescing, and reserve management.

#### 3.3.3. Baselines

For context, we computed mean and persistence (last-value) baselines on the same chronological split; both underperformed compared to the learned models.

### 3.4. Power-Budget Modeling

The average draw is modeled as(1)$$P_{\mathrm{total}}=P_{\mathrm{active}}\cdot DC_{\mathrm{active}}+P_{\mathrm{sleep}}\cdot DC_{\mathrm{sleep}},$$*Notation:*
$P_{\mathrm{active}}$ and $P_{\mathrm{sleep}}$ (mA) denote the average module currents in active and sleep states under the deployed firmware profile; $DC_{\mathrm{active}}$ and $DC_{\mathrm{sleep}}$ are the corresponding duty fractions (with $DC_{\mathrm{active}}+DC_{\mathrm{sleep}}=1$ for a given module). Daily requirements in A·h are obtained by time-weighting and summing across modules over a 24 h mission.

#### Measurement Protocol (Power)

Module currents were bench-measured under the deployed firmware profile. For each module, active and sleep currents were recorded with a precision DMM/data logger (resolution ≤1 mA) at a nominal 12.8 V, first in isolation and then in situ with the controller to confirm consistency. Each measurement covered multiple duty cycles (2 s active, 3 s sleep) and was averaged over repeated runs at a stable lab temperature. These values are represented by $P_{\mathrm{active}}$ and $P_{\mathrm{sleep}}$ in (Equation 1).

Table 1 presents the module-wise daily requirements with a 5 s cycle (2 s active, 3 s sleep) across modules, which balances excursion-detection latency during loading/stops against daily draw to meet the ≥3 day autonomy target (longer 10–30 s cycles reduce draw but may miss short heat spikes; the observed 4-day endurance indicates an adequate margin at 5 s); the total is 24.620 A·h day^−1^. For four days, the capacity is 98.482 A·h, and adding a 30% reserve yields 128.027 A·h. A 4S × 6P LiFePO_4_ configuration (24 cells, nominal 12.8 V, 132 A·h) satisfies the voltage and capacity targets.

## 4. Results

### 4.1. Energy Consumption and Battery Sizing

Applying (Equation 1) and scaling it to four days gave 98.482 A·h; with a 30% reserve, the required pack was 128.027 A·h. A 4S × 6P LiFePO_4_ pack (12.8 V, 132 A·h) met both the nominal voltage and capacity targets in bench trials across multiple four-day runs.

#### Purpose of the AI (Energy Consumption Prediction)

In our system, we use lightweight models (LR, GBM, RF) for *short-horizon energy consumption prediction*. This serves two practical purposes: (i) *verification* of the selected LiFePO_4_ pack against a ≥3-day autonomy target (4 days of autonomy was demonstrated), and (ii) *simplified device configuration* when stress increases (shorter sampling, more frequent burst transmission, and a reserve safeguard). The AI is therefore operational and system-facing rather than a stand-alone ML tool. Across the bench and field trials, the logger completed all four-day runs without brown-out, and the measured depth of discharge remained within the 30% reserve margin established by the analytical budget. In practice, the predictor is used to maintain this safety margin: when the forecast approaches the daily budget, the device switches to a more conservative profile (Section 4.2) so that a one-day reserve is preserved. A detailed comparison of total energy use against a fixed, non-adaptive policy is beyond the scope of the present study, but represents an avenue for future research.

### 4.2. Model Performance and Feature Importance

Across ∼10,000 samples, LR and GBM explained a large share of variance ($R^{2}\approx 0.88$), with the test MAE close to 0.77 A·h. On the same chronological split, the *mean* and *persistence (last-value)* baselines yielded higher errors and lower $R^{2}$ than LR/GBM/RF, confirming that the learned models meaningfully outperform trivial predictors. Residual plots showed no obvious trend of or inflation in variance across the operating range. RF achieved comparable errors while offering interpretable attributions: temperature was the *dominant predictor in the fitted models*, followed by route duration and RH (e.g., ∼55%, 25%, 20%). The multilayer perceptron (MLP) underperforms compared to the linear and tree-based models on this dataset. We attribute this to the relatively small sample size (∼10,000 records across four days) and the low-dimensional, nearly monotonic relationship between the predictors (temperature, RH, elapsed duration, transmission activity) and power draw. In this regime, a more flexible neural architecture offers limited benefits and can overfit without careful regularization, whereas well-tuned linear and ensemble models already capture the dominant trends. Importance is reported as permutation importance on the RF model, and pairwise correlations and variance-inflation checks do not indicate severe multicollinearity among the predictors. These predictions were used only to validate simple adjustments (shorter sampling and more frequent bursts during high temperatures) while keeping a one-day reserve; the primary focus remains on the system’s reliable data acquisition and continuity. We selected LR and GBM for their strong accuracy (test $R^{2}\approx 0.88$, MAE ≈0.77 A·h), and RF for its interpretability; these predictions are used only for sizing verification and simplifying the configuration of the deployed logger. The permutation importance for RF is shown in Figure 6, while Figure 5 reports the MAE/MSE/R^2^ for the five base models (LR, SVM, k-NN, GBM, and MLP); RF achieves MAE and R^2^ values comparable to those of LR and GBM (see above), but is omitted from the bar chart to keep the comparison uncluttered. Figure 7 focuses on RF as the most interpretable ensemble, showing the predicted versus actual energy consumption on the held-out set.

#### Operational Countermeasures (Derived from Predictions)

To maintain a focus on the designed system, we use the energy predictor only to validate simple, actionable configuration of the device. When the consumption over the next 24 h is forecast to exceed 0.8×, the day budget *and* ambient temperature increase, and the logger shortens the sampling interval (e.g., 60 s → 30 s) and increases burst transmit frequency (e.g., 10 min → 5 min) to improve excursion visibility. If the pack is projected to drop below a one-day reserve, the device switches to a low-power profile (e.g., 90 s sampling; deferring non-urgent uplink until coverage becomes available) and raises a dashboard alert that the pack should be swapped at the next hub. If residuals spike (potential anomaly), the current policy is retained and an alert is issued indicating a need for operator inspection. This process demonstrates that the behavior of the system can be simplified without changing the system-first operation.

### 4.3. Field Prototype and Telemetry

The dashboards show continuous T/RH trends, GPS location, and speed data. Figure 8 shows the on-route temperature and relative humidity time series, while Figure 9 presents vehicle speed over the same time window. For clarity, we do not include the raw latitude/longitude dashboard view as a separate figure, because the UI rounds coordinates to two decimal places and normalizes them on a 0–100 scale for fast rendering, which can be visually misleading. As described in Section 3.1, the underlying time series and CSV exports retain the precise coordinates (six decimal places) that are used to compute speed and route duration. Short spikes during lab→vehicle transitions reflect contextual changes rather than sensor drift, and continuity across coverage gaps confirms offline-first logging and authenticated backfill.

## 5. Discussion

This section interprets the system outcomes and clarifies the scope and limitations of this study in relation to typical agri-food transport corridors.

Two outcomes of this study stand out with regard to perishable transport. First, a battery-first co-design enables reliable autonomy *without vehicle power*, and is an adequate size to achieve a ≥3-day autonomy target (*demonstrating 4 days of autonomy* in field trials) while preserving continuous telemetry. In deployment, the device connects via an 802.11 network to a **battery-powered 4G-SIM router** (vehicle-independent uplink): sensing and control are real-time and on-device, and cloud visibility is near-real-time when coverage exists; during gaps, records are buffered and *verifiably backfilled* on reconnection. Multiple sensor nodes can share the same SSID and publish concurrently under separate, device-scoped MQTT topics. Second, lightweight models (LR/GBM/RF) enable short-horizon *prediction of energy consumption*, which is used operationally to (i) *verify* the chosen battery sizing for long domestic corridors and (ii) trigger a simplified logger configuration (shorter sampling, more frequent burst transmission, reserve safeguard) during hot spells or prolonged stops. Together, these address autonomy and visibility on routes with intermittent coverage and close the sensing-to-action loop. This combination of analytical power budgeting and multi-day bench/field trials is the primary energy dimension demonstrated in this study; the ML predictor acts as a guardrail to keep consumption within the planned range and maintain a one-day reserve, rather than to completely minimize absolute energy use. Although recurrent neural architectures, such as long short-term memory (LSTM) or gated recurrent unit (GRU) networks, are natural candidates for richer, multi-step time-series forecasting, this study focuses on short-horizon budgeting from a modest four-day dataset. In this setting, simpler regressors are sufficient to meet the operational target we set (battery sizing and configuration triggers), and more complex sequence models represent an avenue of research for longer studies in the future (see Section 6).

Compared with prior agri-food IoT frameworks, which emphasize traceability but rarely integrate explicit energy modeling for stand-alone, battery-first deployments, this study quantifies the main predictors of power draw *in the fitted models* and translates analytics into straightforward policy (sampling cadence, radio bursts) while maintaining *vehicle-independent* uplink and continuity via opportunistic MQTT and authenticated backfill.

*Threats to validity:* External validity is constrained by the trial window, climate, and mix of goods being transported. Corridor coverage constrains WAN availability, with the battery-powered 4G-SIM router providing no backhaul in low-signal segments, causing updates to be deferred; however, the system remains real-time and on-device and replays buffered data at the next cell/hub. In addition, generalization of the system to refrigerated trucks or multi-modal routes may require re-calibration. Moreover, GPS jitter at very low speeds and intermittent wireless links introduce minor noise, which our preprocessing mitigates but does not eliminate. Finally, our tests were executed in a standard dry truck (non-commercial run) to validate the logger and uplink; while the thermal/coverage conditions reflect typical corridors, the results may differ under live refrigeration loads and handling practices.

## 6. Conclusions and Future Research

We demonstrate the practical application of a stand-alone, vehicle-independent monitoring platform and its operational use of short-horizon energy prediction to inform battery sizing and simplify logger configuration.

We present a vehicle-independent, stand-alone IoT platform for perishable logistics that unifies resilient sensing (T/RH/GPS), energy-aware firmware (deep sleep, burst transmission), and an online/offline pipeline with verified backfill. The system is adequately sized to achieve a ≥3-day autonomy target, *demonstrating 4 days of autonomy* in field trials. Lightweight AI is used primarily for *short-horizon energy consumption prediction* to *verify* battery sizing and to simplify the configuration of the logger (shorter sampling, more frequent burst transmission, reserve safeguard) during hot spells or prolonged stops. LR/GBM achieved strong accuracy (test $R^{2}\approx 0.88$, MAE ≈0.77 A·h), and RF provided interpretable predictors (temperature > duration > humidity) *in the fitted models*, supporting this simple process while keeping the operation system-first.

Future research will (i) extend trials to live commercial shipments, including refrigerated trailers and shipments of multiple goods; (ii) evaluate multi-node deployments to capture spatial gradients within large trucks; (iii) improve security (TLS for MQTT, at-rest encryption) and explore energy harvesting/adaptive sleep to widen autonomy margins; and (iv) broaden datasets to encompass different seasons and corridors, including sequence-aware predictors (e.g., LSTM/GRU architectures) for route context and lightweight anomaly detection.

## Figures and Tables

**Figure 1 sensors-25-07475-f001:**
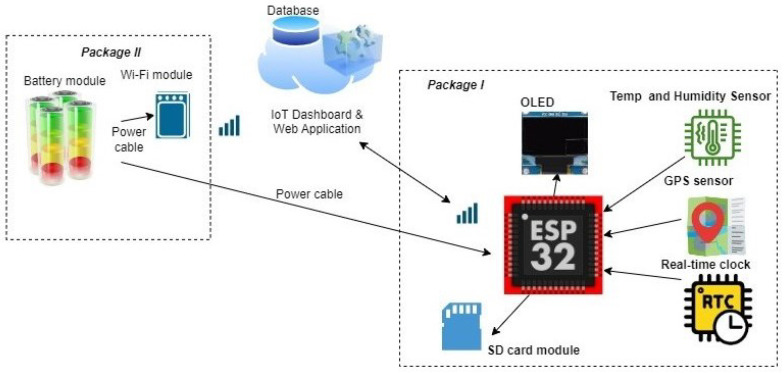
The system architecture.

**Figure 2 sensors-25-07475-f002:**
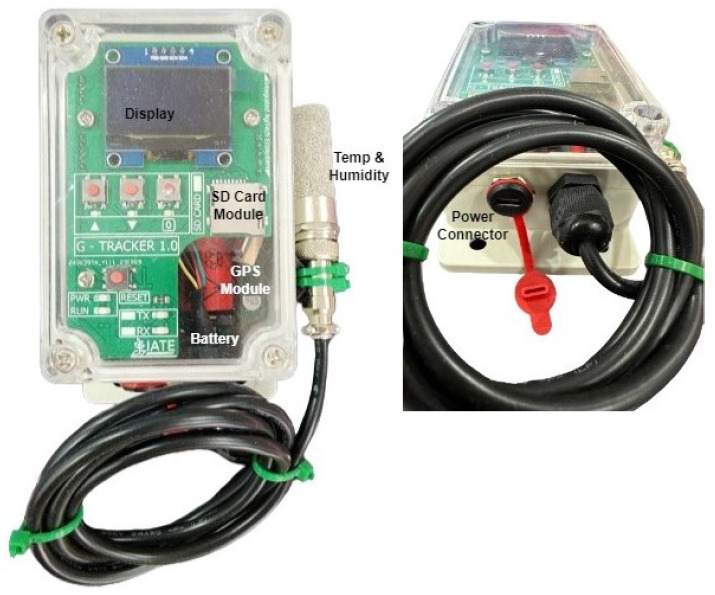
The system controller module (packet I).

**Figure 3 sensors-25-07475-f003:**
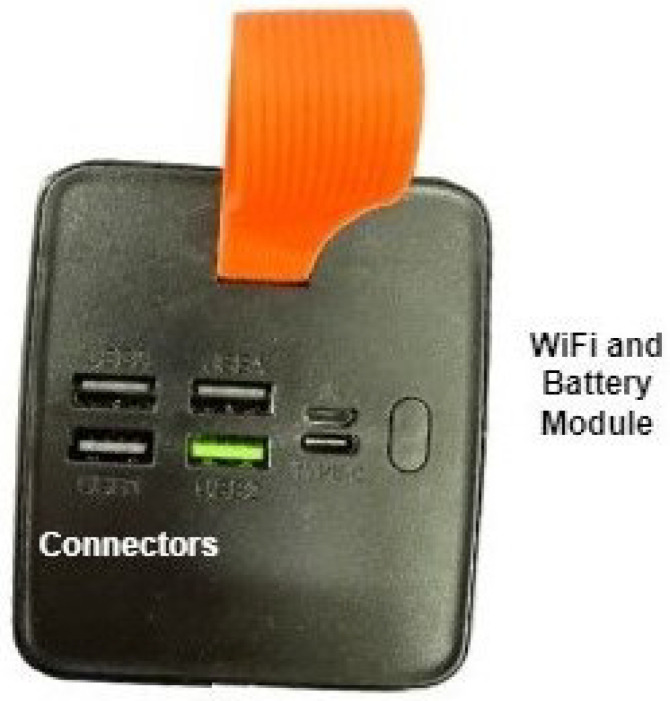
The power supply and internet module (packet II).

**Figure 4 sensors-25-07475-f004:**
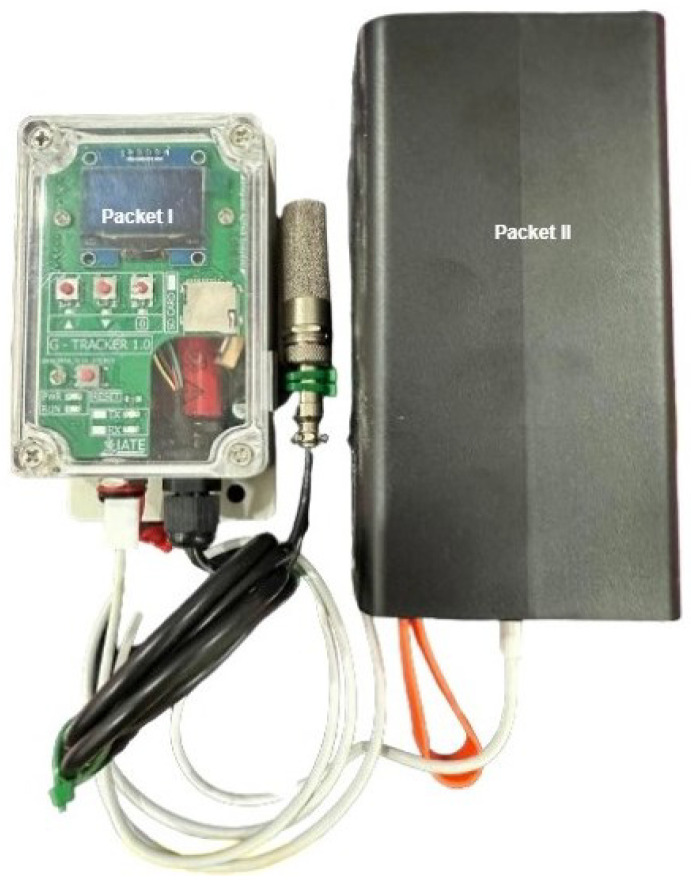
The connection between packets I and II of the ITEMS system.

**Figure 5 sensors-25-07475-f005:**
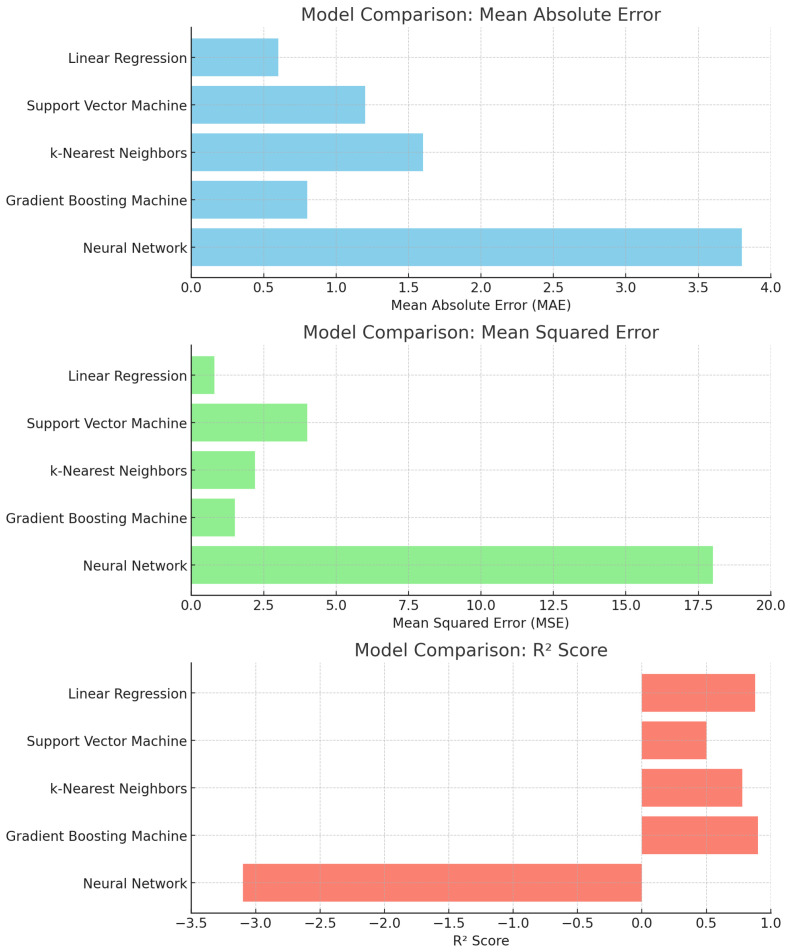
Model performance comparison (MAE, MSE, R^2^) across the five base models: LR, SVM, k-NN, GBM, and MLP (Neural Network). Random Forest (RF) obtains an MAE and R^2^ comparable to those of LR/GBM, and is discussed separately in Section 4.2 and Figure 6 and Figure 7.

**Figure 6 sensors-25-07475-f006:**
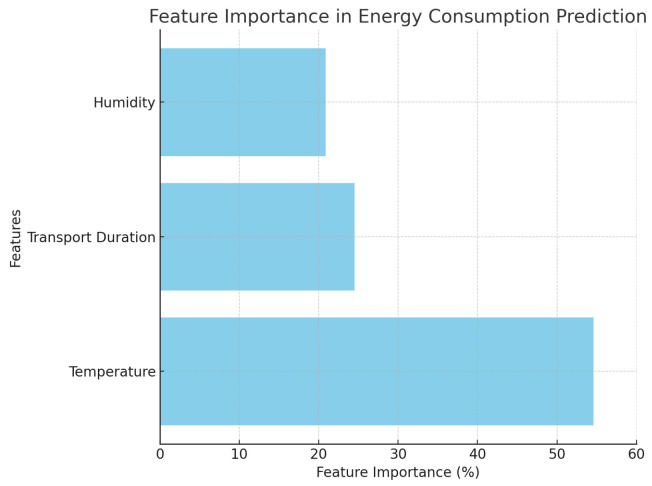
Feature importance (Random Forest) for predicting short-horizon energy consumption. Bars show normalized importance; temperature dominates, followed by transport duration and RH.

**Figure 7 sensors-25-07475-f007:**
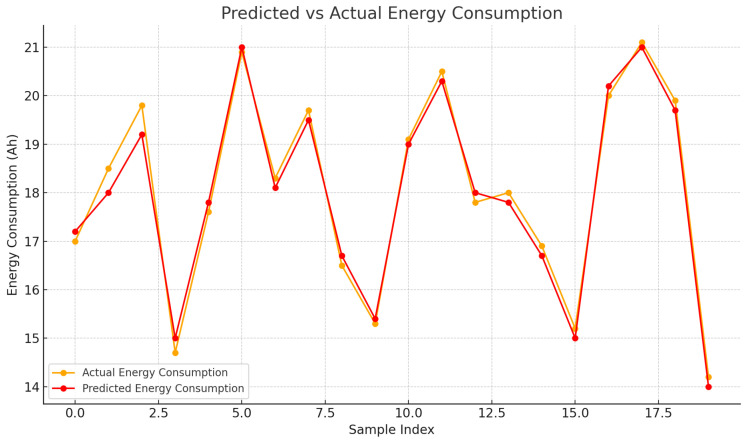
Predicted vs. actual energy consumption on the held-out set (RF).

**Figure 8 sensors-25-07475-f008:**
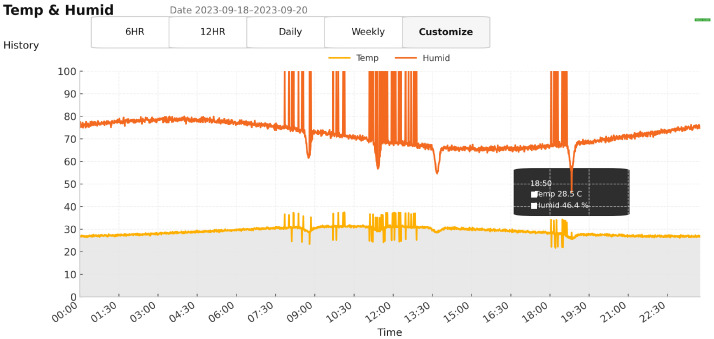
Real-time temperature and RH trends during transport.

**Figure 9 sensors-25-07475-f009:**
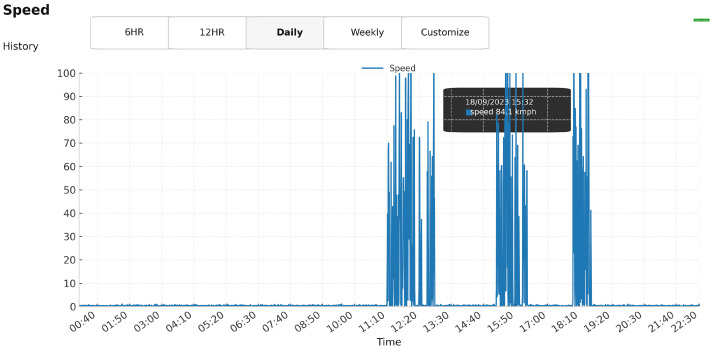
Vehicle speed along the route. Clusters correspond to in-motion segments separated by brief stops.

**Table 1 sensors-25-07475-t001:** Module’s daily power requirement (24 h mission, 5 s cycles).

Module	Power (A·h/day)
ESP32 (active)	10.368
ESP32 (standby)	1.555
Temp/RH sensor (active)	0.1728
Temp/RH sensor (standby)	0.031
GPS (active)	1.728
GPS (standby)	0.2592
microSD (active)	3.456
microSD (standby)	0.2592
RTC	0.1728
Wi-Fi module (active)	6.912
Wi-Fi module (standby)	0.155
**Total (per day)**	**24.620**

## Data Availability

The dataset and firmware settings used in this study are available from the corresponding author upon reasonable request.

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
