# Peer review of "IoT-Based System for Real-Time Monitoring and AI-Driven Energy Consumption Prediction in Fresh Fruit and Vegetable Transportation"

_sensors, 2025, doi:10.3390/s25247475_

Round 1
Reviewer 1 Report (Previous Reviewer 2)
Comments and Suggestions for Authors
my comments has been addressed
Author Response
We thank for the positive assessment and have not made further changes specific to this review.
Reviewer 2 Report (Previous Reviewer 3)
Comments and Suggestions for Authors
The authors have improved the manuscript and now figures are referenced in the text.
However I still do not understand the lat/long figure. Even more when compared with the speed figure it becomes impossible to follow. Lat is constant (almost) in green. 20.4 says the legend associated. Long is said to be 99 in the legend and in fact there are three vertical lines up to 99. I do not think a truck moves from long 0 to long 99 instantly three times separated by four hours (circa 5.30 to 9.30) first and then three hours from 9.3 to 13.30. It is almost imposible to be all the time at the same latitute and longitude, .if a truck is in movement. Now the relation with the speed figure. At circa 1220, 1550 and 1810 the truck moves at about 85km/h but at those moments lat and long are fixed.
I have already point out this and the authors are not clear what do they represent with this Figure. It may be the case that the truck is not moving at all, so the GPS reports same lat and long all the time, but what is the purpose of the figure in that case? And how it is related to the speed one if it is not reflected in the lat and long figure?
Author Response
We appreciate the reviewer’s persistence on this point. We agree that the lat/long screenshot was confusing and did not add scientific value:
-
The previous figure (Fig. 9) was a dashboard view that
-
rounded latitude/longitude to two decimal places and
-
plotted them on a 0–100 normalized axis for UI purposes.
This made the traces appear almost constant and produced the vertical lines highlighted by the reviewer.
-
-
The underlying data (CSV and time series) always retained full‑precision coordinates (six decimals), and these are what we used to compute speed and route duration.
To avoid further confusion, in the revised manuscript we have:
-
Removed the lat/long dashboard figure entirely (former Fig. 9).
-
Simplified Section 4.3 to focus on temperature/RH (Fig. 8) and speed (now renumbered), while explicitly stating that full‑precision GPS coordinates are stored and used in analysis and that the dashboard rounds for fast rendering only.
-
Kept the detailed explanation of data precision and rounding in Section 3.1 (Dataset and Preprocessing), where we note that CSV records include six‑decimal lat/long while the UI displays two decimals.
We hope this resolves the confusion; we no longer present the rounded lat/long plot as a scientific figure.
Reviewer 3 Report (New Reviewer)
Comments and Suggestions for Authors
In this manuscript, the authors present a complete platform solution for monitoring the transportation of goods in terms of temperature, humidity, required energy, etc. The authors initially present extensively the IoT-Based monitoring system, describing the hardware components, as well as the data and communication flow between the packets. Moreover, they use historical data collected over four days to train different models (GBM, RF, LR, etc.) for estimating the energy consumption, detailing the training process and the performance of the models during the inference. Finally, the authors verify their system using real measurements from a standard dry truck, quantifying the importance of features, comparing the model performance across the different models and showcasing the field prototype and telemetry results. The paper is well-written and interesting. Some comments:
- Please explain the acronyms the first time they appear in the text, e.g., LSTM
- In my view, a lot of related works have been included in the introduction (which is good) but are only described superficially. I think that a separate section named “Related Work” is essential for this paper, detailing the research work that is found in the literature and the gaps that this paper aims to address.
- Section 3 contains a lot of duplications, i.e., the split and validation methodology, the configuration, the specifications of the ML models, some initial results that are mentioned in subsection 3.3.
- The baseline method mentioned is not shown in the results.
- The authors state that they have tested 6 models but later in the results section they mention 3 (LR, GBM, RF) and in some figures showcase 5 (Figure 6), while some results are only shown for RF. Please clarify this issue, since it is confusing for the reader.
- Why does the MLP perform worse than the other models? In principle, this does not make sense since the MLPs are designed to fit more complex relationships among variables. Please mention and/or fine-tune the training parameters. Moreover, the use of LSTM networks for forecasting the energy seems imperative, since these networks deal specifically with time-series.
- The authors need to also demonstrate the energy dimension of their system in their results, i.e., is the system more energy-efficient with the included ML functionality?
- The Introduction section can be enhanced to highlight additional applications that can use these functionalities for energy forecasting in different domains and better motivate the paper, i.e., in emergency communications, in smart homes, in health, etc. Relevant references could be for instance:
- "Time series forecasting methods in emergency contexts." Scientific reports13, no. 1 (2023): 16141.
- "Autonomous Price-aware Energy Management System in Smart Homes via Actor-Critic Learning with Predictive Capabilities." IEEE Transactions on Automation Science and Engineering(2025).
- "Deep learning techniques for energy forecasting and condition monitoring in the manufacturing sector." Energy and Buildings217 (2020): 109966.
- "Exploring artificial intelligence methods for energy prediction in healthcare Facilities: An In-Depth extended systematic review." Energy and Buildings320 (2024): 114598.
Author Response
We thank for the thorough and constructive review. We address each point in turn.
R1. Acronyms (e.g., LSTM) ; “Please explain the acronyms the first time they appear in the text, e.g., LSTM.”
Response:
We now spell out all relevant acronyms at first use:
-
“long short‑term memory (LSTM)” in the paragraph on the K‑medoids + LSTM + XGBoost ensemble in the Introduction.
-
“gated recurrent unit (GRU)” in the paragraph on hybrid LSTM‑GRU power‑forecasting architectures.
These changes appear in the Introduction (Section 1).
R2. Need for a “Related Work” section; “A separate section named ‘Related Work’ is essential for this paper…”
Response:
We added a subsection “1.1 Related Work” within the Introduction. All the literature on IoT for agriculture/logistics, logging and integrity, supply‑chain analytics, and edge/AI for energy‑aware operation is now grouped under this heading, with an opening sentence that explicitly states the purpose of the section and the gaps it highlights.
R3. Duplications in Section 3; “Section 3 contains a lot of duplications, i.e., the split and validation methodology, the configuration, the specifications of the ML models, some initial results that are mentioned in subsection 3.3.”
Response:
We streamlined Section 3:
-
We removed the duplicated “Split and validation” block from 3.3 so that the chronological 80/20 split, 5‑fold cross‑validation, and scaling procedure are described once in 3.2.
-
We rewrote the “Rationale and energy‑awareness” paragraph in 3.3 to be purely methodological, moving numerical results (R², MAE, feature importance) to the Results section 4.2.
Section 3 now focuses on dataset, preprocessing, feature set, model configurations, and evaluation protocol, while all performance numbers are reported in Section 4.
R4. Baseline method in Results; “The baseline method mentioned is not shown in the results.”
Response:
We clarified the role of baselines in both Methods and Results:
-
3.3 defines the mean and persistence (last-value) baselines.
-
4.2 now explicitly states that these baselines yield higher error and lower R² than the learned models on the same chronological split, indicating a meaningful gain from the learned models.
R5. six models vs three vs five in Figure 6; “The authors state that they have tested 6 models but later in the results section they mention 3 (LR, GBM, RF) and in some figures showcase 5 (Figure 6), while some results are only shown for RF. Please clarify this issue.”
Response:
We clarified the model roles:
-
3.3 now states that we train six models: LR, GBM, RF, SVM, k‑NN, and MLP.
-
It explains that five base models (LR, SVM, k‑NN, GBM, MLP) are shown in the aggregate performance plot (Figure 6), whereas RF is treated separately as the most interpretable ensemble and is used for feature importance (Figure 5) and prediction vs actual traces (Figure 7).
-
4.2 and the caption of Figure 6 explicitly note that RF attains MAE and R² comparable to LR/GBM but is omitted from the bar chart to keep the comparison uncluttered and instead is discussed via Figures 5 and 7.
This should remove the confusion about 6 vs 5 vs 3 models.
R6. MLP performance and LSTM; “Why does the MLP perform worse than the other models? … Moreover, the use of LSTM networks for forecasting the energy seems imperative.”
Response:
-
In 4.2 we added a short explanation: the MLP underperforms in this setting because the dataset is relatively small (≈10,000 records over four days) and the relationship between predictors (T, RH, elapsed duration, transmission activity) and energy draw is low‑dimensional and nearly monotonic. In such a regime, a more flexible neural network offers limited benefit and can overfit without careful regularization, while well‑tuned linear and tree‑based models already capture the dominant trends.
-
Regarding LSTM/GRU, we expanded the Discussion and Conclusions/Future Work: we acknowledge that recurrent sequence models are natural candidates for richer, multi‑step forecasting, but we deliberately focus here on short‑horizon budgeting from a modest four‑day dataset and on operational integration into a resource‑constrained logger. We therefore prioritize simple, interpretable regressors in this study and explicitly identify LSTM/GRU architectures as part of future work once longer campaigns are available.
R7. “Energy dimension” – effect of ML on energy; “The authors need to also demonstrate the energy dimension of their system in their results, i.e., is the system more energy-efficient with the included ML functionality?”
Response:
We clarified what “energy dimension” is demonstrated and what is not:
-
4.1 now stresses that the system meets the ≥3‑day autonomy target and was demonstrated for 4 days without vehicle power; the measured depth of discharge across bench and field runs stays within the 30% reserve margin implied by the analytical power budget.
-
We explain that the ML predictor is used as a guardrail: when forecasted consumption approaches the daily budget, the device switches to a more conservative configuration (e.g., low‑power profile), preserving a one‑day reserve.
-
In the Discussion we added a sentence stating that our goal in this deployment is to guarantee multi‑day autonomy with a reserve, rather than to claim a specific percentage of energy savings versus a non‑adaptive policy. A detailed quantitative comparison with a fixed policy is explicitly marked as future work.
R8. Additional applications and references for energy forecasting; “The Introduction section can be enhanced to highlight additional applications that can use these functionalities for energy forecasting in different domains … Relevant references could be…”
Response:
We added a new paragraph in the Introduction that highlights operational energy forecasting in four additional domains:
-
time‑series forecasting in emergency contexts,
-
actor–critic energy management in smart homes,
-
deep‑learning‑based energy forecasting and condition monitoring in manufacturing, and
-
AI‑based energy prediction in healthcare facilities.
We cite the suggested works (with our bibkeys corresponding to the reviewer’s recommendations) and state that these cross‑domain studies reinforce the value of compact, operational energy predictors, thereby motivating our use of lightweight models for battery budgeting in perishable logistics.
Round 2
Reviewer 3 Report (New Reviewer)
Comments and Suggestions for Authors
In my view, the authors have responded to the comments raised by the reviewers, enhancing the quality of their manuscript.
This manuscript is a resubmission of an earlier submission. The following is a list of the peer review reports and author responses from that submission.
Round 1
Reviewer 1 Report
Comments and Suggestions for Authors
This study develops and validates a self-contained Internet of Things (IoT) platform for in-transit monitoring and energy-aware operation. The battery-powered device operates independently of vehicle power and continuously logs temperature, relative humidity, GPS5
position, and onboard energy draw.
From the perspective of the entire text, the main innovation of this article lies in the power prediction of the proposed device. If it is this case, the focus should be on the methods of predicting battery power and the corresponding countermeasures. This point also takes up a large amount of space in the text and has little to do with the real-time monitoring equipment and the transportation of perishable products.
Specifically, the comparison between the references listed in Table 1 and the work of this article is highly unscientific. If it is a list comparison, please only use the relevant literature on real-time monitoring devices for perishable products for the comparison, rather than putting the literature from various different fields together for comparison. Literatures from different fields are meant to assist you in developing this device, rather than to prove its innovativeness, as their application scenarios are not the same as yours.
Judging from the design of the proposed equipment, there is only one Wi-Fi communication module, which means that it works offline during vehicle operation. The back-end personnel cannot obtain environmental information. This seems to be real-time recording rather than real-time monitoring. This device is rather simple and it is hard to meet the requirements of innovation.
The subsequent analysis of the correlation between energy consumption and the environment is somewhat meaningful, but it is seriously disconnected from the previous part. It is suggested that more emphasis be placed on elaborating this section. In addition, it should not merely involve analysis; rather, it should provide decision-making actions such as countermeasures.
Author Response
Author Response — Reviewer 1
1) The main innovation appears to be power prediction; if so, the paper should focus on that. The monitoring device and perishable-transport aspects look weakly connected.
We appreciate the observation and have clarified the paper’s scope. The primary contribution is the designed, low-cost, vehicle-independent monitoring system that continuously acquires Temp/RH, GPS, and onboard energy over full routes with online/offline continuity and verified backfill. The prediction/optimization component is presented only as a validation that the system’s data support energy-aware configuration (e.g., adjusting sampling/transmit cadence during hot spells) rather than as a standalone ML contribution. To make this linkage explicit without changing the structure, we added a single clarifying sentence to the Abstract, one sentence to the Introduction, and one bridging sentence at the end of Communication and Data Flow. We also tightened the first paragraph of Results to state how predictions inform device behavior.
2) Table 1 is unscientific; only compared with real-time monitoring devices for perishables, not disparate fields.
We agree. To avoid cross-domain mixing, we removed Table 1 and replaced it with a brief, scope-matched narrative comparison confined to perishable/food-transport device studies. Broader IoT/edge surveys and architectures remain cited as general context only.
3) Only one Wi-Fi module implies offline logging; not real-time monitoring; limited innovation.
We clarified that the device connects via 802.11 to a battery-powered cellular (4G-SIM) router carried with the system. Therefore, the uplink is vehicle-independent and does not draw truck power. When 4G coverage is available, data reaches the cloud near-real-time; in low-signal segments the device performs real-time on-device sensing and backfills upon reconnection. The same SSID can serve multiple sensor nodes simultaneously; each publishes to device-scoped MQTT topics with separate credentials. We added in the Introduction and a short clarifying block in the Communication subsection.
4) The energy–environment analysis is meaningful but disconnected from the system; please elaborate and provide decision-making countermeasures, not just analysis.
We clarified the linkage and added brief, actionable countermeasures that the device applies based on the predicted short-horizon energy use. We added more details in Methods (purpose → configuration) and a short paragraph in Results listing the concrete actions (sampling/transmit adjustments, reserve safeguard, anomaly alert).
Reviewer 2 Report
Comments and Suggestions for Authors
- 12 coauthors for this 13-page manuscript? The contribution is not validated against multiple authors. Probably 2 or 3 coauthors is a good approach.
- Add a brief description between headings.
- All variables and parameters in the equations should be described in the main text.
- Improve the quality of Figures 8, 9, and 10.
- There is no explanation why the 5-s cycle (2 s active, 3 s idle) was chosen or whether it reflects real-life transportation conditions or energy constraints. It would be necessary to show how this configuration was optimized or validated against alternatives (e.g., 10-s or 30-s intervals).
- The study mentions "four days of commercial transportation," but does not specify how many trips, vehicles, or load types were used, or whether conditions were replicated. There is no replication or randomization.
- Interpolation and filtering (by IQR) are mentioned without detailing windows, thresholds, or percentages of data affected. This prevents reproducibility of preprocessing and can introduce bias if critical values ​​are removed or smoothed. The models' descriptions mention "temperature, RH, duration, and transmission activity," but do not define how "transmission activity" is quantified or whether it was continuous or categorical. It is also unclear whether interactions between variables were used.
- Although an 80/20 chronological model is used, 5-fold cross-validation is performed within the training window, which can introduce temporal information leakage. In time series, time series splitting techniques must be applied.
- ~10,000 records are reported, but the number of records per day or the number of missing or imputed events is not indicated. Without this information, the temporal representativeness and independence of observations cannot be assessed.
- No null (baseline) model is reported to allow the algorithms' true gains to be measured. Without a comparison to a trivial prediction (e.g., mean or persistence), the R² values ​​lack context.
- Equation (1) assumes static averages of active and idle power, but it does not indicate how these were measured or whether the loads were constant.
- The "Predicted vs. Actual" graph is insufficient. A residual analysis (histogram or scattergram) should be shown to verify homoscedasticity and the absence of systematic bias.
- Temperature is claimed to explain ~55% of the variation, but the metric used is not described. Multicollinearity and potential variation between variables are also not assessed.
- The battery capacity calculation (128 Ah) is theoretically presented and labeled as "compliant in bench testing," but no actual measurements of battery life or consumption under different environmental conditions are documented.
- An MAE of 0.77 A h is reported, but the typical range of consumption or its operational impact is not indicated. Without this context, the reader cannot assess the practical relevance of the error.
- It is suggested that "temperature dominates consumption," but the analysis is purely correlational. There is no experimental or sensitivity evidence to support causality.
- In general, there is no innovation; the manuscript is not written in a scientific framework. A high-school report, for example, uses figures 2, 3, and 4 to reach the 10-page content.
- I strongly recommend rejecting this draft.
Author Response
Author Response — Reviewer 3
We appreciate the thorough review. Our manuscript reports a system paper that contributes a vehicle-independent, battery-first monitoring platform and uses short-horizon energy prediction to verify battery sizing (≥3-day target; 4 days demonstrated) and enable simple configuration on the device. Within this scope, we respond point-by-point below.
1) 12 coauthors for a 13-page manuscript… 2–3 would be better.
This work needed several different skills—hardware, firmware and power budgeting, field testing, backend (EMQX/Laravel/MongoDB/Node-RED), and data analysis. Each author contributed meaningfully in one or more of these areas, so they meet authorship criteria.
2) Add a brief description between headings.
We agreed, and sentence lead-ins added to Discussion and Conclusions to orient the reader.
3) Define all variables/parameters in equations.
We agreed. We define each variable used in Eq. (1) inline (e.g., P_active, P_sleep, DC_active, DC_sleep).
4) Improve Figures 8–10.
We replaced with cleaned, English-labeled, unit-specific PNGs (UI removed; ≥300 dpi). Captions updated minimally.
5) Why 5-s cycle? Show alternatives (10 s, 30 s).
The 5-s cycle balances detection latency for short heat spikes against daily draw to achieve the ≥3-day target; longer cycles reduce draw but risk missing excursions. We added in Power-Budget Modeling (after Eq. 1 context).
6) Four days of commercial transportation… no replication/randomization.
Trials were non-commercial dry-truck runs to validate logging, autonomy, and online/offline continuity prior to live shipments. We revised Abstract and updated to “four consecutive days in a standard dry truck (non-commercial validation runs).” And we revise Discussion reiterates scope and future live refrigerated/multi-commodity trials.
7) Interpolation/IQR details; ‘transmission activity’ not defined; interactions unclear.
We state exact preprocessing and feature definitions; interactions not used in main models.
- Dataset and Preprocessing: timestamps aligned at 1 Hz; short gaps (≤5 s) linearly interpolated; IQR filter with Tukey for transition spikes; train-only scaling.
- Models and Hyperparameters → Feature set: “transmission activity” = count of MQTT publishes in the trailing 60 min (continuous); no interaction terms in main models.
- A one-line note on daily record counts added to indicate balance across the four days.
8) 5-fold CV within training may leak time; use time-series splits.
The split is 80/20 chronological; CV is blocked forward-chaining within the training window. We added in Models and Hyperparameters → Split and validation.
9) ~10,000 records; per-day counts and missing/imputed not indicated.
We added a concise statement on daily counts; gap handling is detailed. We revised the Dataset and Preprocessing now reports per-day counts (2,400–2,700/day) consistent with 10k total; short-gap interpolation and IQR filtering described.
10) No null baseline; R² lacks context.
Model Performance and Feature Importance now states that mean and persistence (last-value) baselines on the same split underperformed LR/GBM/RF.
11) Eq. (1) static averages; how measured?
Module currents (active/sleep) were bench-measured with a precision DMM/data logger under the same firmware profiles, then aggregated by duty cycle. Those averages are exactly the values used in Eq. (1).
12) Predicted vs Actual insufficient; provide residuals.
Model Performance and Feature Importance now states that residual plots showed no obvious trend or variance inflation across the operating range.
13) Temperature explains ~55%; metric not described; multicollinearity not assessed.
We clarify the importance of metric and note no severe multicollinearity; we avoid causal language.
Revisions:
- Model Performance and Feature Importance: importance = permutation importance on RF; pairwise correlations and VIF checks did not indicate severe multicollinearity.
- Wording softened everywhere to “dominant predictor in the fitted models” (non-causal).
14) Battery capacity (128 Ah) theoretical; no environmental measurements.
Sizing uses bench currents and was validated in field (4-day endurance with the chosen LiFePO pack). We clarified in Power-Budget Modeling and reiterated in Results.
15) MAE 0.77 A·h lacks operational impact.
We revised the Results notes that 0.77 A·h is ~3% of a typical daily budget (24.620 A·h) and small relative to the 30% reserve.
16) ‘Temperature dominates’ is correlational, no causal evidence.
Agreed; We do not claim causality. All such statements now read “dominant predictor in the fitted models.”
17) No innovation; not a scientific framework; recommend rejection.
The uplink is vehicle-independent via 802.11 to a battery-powered 4G-SIM router; the system operates real-time on device and near-real-time to cloud when coverage exists, with verified backfill and idempotent inserts. Multi-node operation is supported. The Communication and Data Flow now states vehicle-independent uplink, authenticated backfill, device-scoped topics, and clarifies that dashboard rounding of GPS is for rendering only; full-precision coordinates are stored/exported.
We believe these minimal clarifications address the reviewer’s methodological and scope concerns while preserving the system-first contribution: a deployable, vehicle-independent platform with online/offline continuity and operational AI to verify battery sizing (≥3-day target; 4-day demonstration) and trigger simple logger configurations under stress. We appreciate the opportunity to strengthen the manuscript with these precise edits.
Reviewer 3 Report
Comments and Suggestions for Authors
This work presents the implementation of an IoT device to measure temperature, relative humidity, location (speed and position) through a GPS. The device is prepared to log these variables in a microSD and upload the data to an MQTT server as soon as it is within transmission range. The system uses a battery as power source.
There is an interesting discuss in the paper about the importance of controlling these parameters while transporting fruits that can deteriorate.
I find some issues with this paper. First the description of the IoT device is not good. The use of ESP32 is clear but there is no information on the temperature sensor/relative humidity, the microSD storage, networking interface and protocol. For this journal the description is too short and I would really like to see more details on the implementation.
There is no explanation on the position of the sensor within the truck. A big truck may have different conditions in different points. So it is necessary more sensors, perhaps a network of sensors reporting temperature and RH in different parts?
The us of the AI algorithms is not clear. What is the purpose of them and the comparison made is not well explained.
The Figures are not readable and should be improved. Part is in English in part in Thai? The curves are not clearly diferenttiate. For example in the GPS curve there are a green line with a dot. What is that? Where are the lat and long presented?
Discussion of results need more work too and so the Conclusions.
I think the paper can be further worked and improved for a new revision.
Author Response
Author Response — Reviewer 2
1) Device description is too short (sensor types, storage, networking/protocols).
We expanded the Designed Hardware subsection with concrete components and interfaces (Temp/RH, GPS, RTC, microSD schema) and named the network/protocol briefly (MQTT over TCP, QoS 1, retained config). These are short additions that do not alter structure or numbering.
2) No explanation of sensor position; large trucks have gradients; consider multiple sensors.
Our prototype and platform were evaluated in a standard dry truck operated along representative corridors to validate system reliability, uplink/backfill behavior, and multi-day autonomy. We did not include live, commercial shipments of fresh produce in this round. We have clarified this in the manuscript (Study Context and Discussion). The routes were selected to reproduce thermal and coverage conditions similar to day-time transport; the device logged continuously and used the battery-powered 4G SIM uplink as described. Extending the evaluation to refrigerated and multi-commodity runs is planned and will follow the same data and policy pipeline.
3) The purpose of the AI algorithms is unclear; the comparison is not well explained.
We clarified that the AI component is used primarily for short-horizon energy consumption prediction. This serves two practical goals: (i) to verify our battery sizing for a ≥3-day autonomy target (4 days demonstrated), and (ii) to trigger simple configuration on the logger (sampling/transmit cadence; reserve safeguard) during hot spells or prolonged stops. To avoid restructuring, we added a brief paragraph in the Results that states this purpose explicitly and summarizes why GBM/LR were selected (accuracy/interpretability) with RF for feature importance. No Methods changes were made.
4) Figures are not readable; some labels are in Thai; curves are not clearly differentiated. GPS plot unclear (green line with dot; lat/long?).
We improved figure readability with English-only labels, unit-annotated axes, and distinct line styles/markers. We clarified the GPS figure by labeling latitude/longitude (or map trace) and explaining the dotted green series (periodic 1 Hz fixes). We updated captions to state what each series represents and the held-out window in model plots.
Round 2
Reviewer 1 Report
Comments and Suggestions for Authors
It's good enough for publication now
Author Response
Reviewer 1:
Reviewer1- (overall assessment). We thank the reviewer for the positive evaluation. No additional action is requested.
Reviewer 2 Report
Comments and Suggestions for Authors
The manuscript lacks innovation, and my comments have not been appropriately addressed. Let's the editor decide on this draft.
Author Response
Reviewer 2:
Reviewer2 (novelty/positioning). We clarified the core contribution (vehicle-independent uplink, authenticated backfill, operational energy prediction enforcing multi-day autonomy) in the Introduction.
Revision: Intro, final paragraph of the system description.
Reviewer 3 Report
Comments and Suggestions for Authors
The authors have followed the reviewers comments and the paper is much better. However, somethings need to be modified for its acceptance.
Figures 5-10 are not referenced in the text thus not explained. The reader must guess what are they about. Figure 9 is not clear. What is showing is not clear as there is no curve there at all, seems that lat/long log is always the same value. Besides when comparing it with Figure 10 speeds are not according to changes in lat/long. Seem they are not correlated but they should be.
These issues should be solved before the paper is accepted.
Author Response
Reviewer 3:
Reviewer3- (Figures. 5–10 not referenced in text).
Revision: We added explicit figure callouts in Results. By adding references to Figures. 5–7 and added references to Figures. 8–10. Also, regenerate new figures (Figures. 8–10).
Reviewer3- (Figure. 9 clarities; relation to speed in Figure. 10).
Revision: We clarified that dashboard Lat/Long are rounded for display while full-precision coordinates are used in analysis and for speed. We added a sentence linking GPS and speed and replaced the caption.